# Effect of Steam Flash-Explosion on Physicochemical Properties and Structure of High-Temperature Denatured Defatted Rice Bran Protein Isolate

**DOI:** 10.3390/molecules28020643

**Published:** 2023-01-08

**Authors:** Zhiguo Na, Haixin Bi, Yingbin Wang, Yujuan Guo, Yongqiang Ma

**Affiliations:** 1School of Food Engineering, Harbin University of Commerce, Harbin 150028, China; 2College of Food Engineering, East University of Heilongjiang, Harbin 150060, China

**Keywords:** high-temperature denatured defatted rice bran, steam flash-explosion, rice bran protein isolate, structural properties

## Abstract

The effects of Steam Flash-Explosion (SFE) on the physicochemical properties and molecular structure of high-temperature denatured defatted rice bran protein isolate (RBPI) were investigated. The mechanism of SFE treatment on high-temperature denatured defatted RBPI was revealed. The analysis of the physical and chemical properties of RBPI showed that the surface hydrophobicity, characteristic viscosity, and thermal stability of rice bran protein isolate were significantly affected by the pressure of saturated steam and pressure holding time. Under the conditions of 2.1 MPa and 210 s, the surface hydrophobicity index decreased significantly from 137.5 to 17.5, and the characteristic viscosity increased significantly. The peak temperature of denaturation decreases from 114.2 to 106.7 °C, and the enthalpy of denaturation decreases from 356.3 to 231.4 J/g. The higher structure (circular dichroic spectrum and endogenous fluorescence spectrum) of rice bran protein isolate was analyzed by volume rejection chromatography (SEC). The results showed that steam flash treatment could depolymerize and aggregate RBPI, and the relative molecular weight distribution changed greatly. The decrease in small molecules with poor solubility was accompanied by the increase in macromolecules (>550 kDa) soluble aggregates, which were the products of a Maillard reaction. The contents of free sulfhydryl and disulfide bonds in high-temperature rice bran meal protein isolate were significantly increased, which resulted in the increase in soluble aggregates containing disulfide bonds. Circular dichroism (CD) analysis showed that the α-helix content of the isolated protein was significantly decreased, the random curl content was increased, and the secondary structure of the isolated protein changed from order to disorder. The results of endogenous fluorescence spectroscopy showed that the high-temperature rice bran meal protein isolate was more extended, tryptophan was in a more hydrophilic microenvironment, the fluorescence intensity was reduced, and the tertiary structure was changed. In addition, the mean particle size and net surface charge of protein isolate increased in the aqueous solution, which was conducive to the development of the functional properties of the protein.

## 1. Introduction

Rice bran, as the main by-product of rice production, is extremely rich in resources. It not only has a huge yield and low price but also concentrates more than 60% of the nutrients of rice [1], has a high content of unsaturated fatty acids such as linoleic acid, up to 80%, and is rich in dozens of natural bioactive substances such as vitamin E, tocopherol trienes and squalene [2]. Compared with other plant proteins, the lysine content in rice bran protein is higher, which not only requires complete amino acids, reasonable composition, close to the FAO/WHO recommended model, and high biological titer but also has the advantages of being hypoallergenic, which is a plant protein resource with great potential for development [3]. The protein content is about 11.3–14.9%. Soluble protein accounted for about 70% of the total, containing all four proteins in Osbron’s classification, namely albumin, globulin, gluten, and glycolysin [4], with a nutritional value comparable to that of egg protein [5]. However, these proteins have strong polymerization and a large number of disulfide bonds, and the high content of dietary fiber and phytic acid in rice bran aggravates the intercoalescence of rice bran components and the functional properties such as solubility decline [6], which makes the extraction and separation of proteins difficult and greatly limits its application in the food industry.

The Steam flash-explosion (SFE) treatment can cause the material to be subjected to the mechanical shear caused by high-temperature cooking and steam release at the same time, which can not only damage the cell wall structure of rice bran but also degrade hemicellulose and release the protein embedded in it. In addition, these effects can also cause changes in molecular structure and physical and chemical properties of proteins in high-temperature denatured defatted rice bran (HTDDRB), thus affecting functional properties such as solubility [7] and effectively improving the extraction rate and nitrogen solubility index of proteins in HTDDRB. In addition, according to the principle and characteristics of the Maillard reaction, high-temperature cooking in the steam flash process may cause covalent binding of proteins and sugars in high-temperature rice bran meal, resulting in increased polarity of protein molecules and changes in spatial structure, leading to changes in solubility [8]. SFE technology, as a typical physicochemical treatment technology, is a treatment method in which raw materials are placed in a closed environment of high-temperature liquid water and high-pressure steam for a certain period of time and then immediately released to atmospheric pressure. It is economical, efficient, and pollution-free [9,10]. At present, there are few reports on the effect of steam flash detonation treatment on the physicochemical properties and structure of rice bran protein isolate.

Therefore, on the basis of previous studies, taking high-temperature rice bran meal protein isolates with high nitrogen dissolution index and high protein extraction rate as the research object, this study analyzed the effects of SFE treatment on the physicochemical properties and structure of high-temperature denatured defatted rice bran protein isolate, and discussed its mechanism of action, providing a theoretical basis for the development and utilization of high-temperature rice bran protein.

## 2. Results and Discussion

### 2.1. Surface Hydrophobicity (H_0_) Analysis

Surface hydrophobicity is an important physical and chemical property of proteins, which is of great significance for protein stability, conformation, and function [11]. The interaction of proteins with water and other compounds in solution is closely related to surface hydrophobicity, so the solubility, emulsification, water absorption, and other functional properties of proteins are affected by surface hydrophobicity. The surface hydrophobicity is also affected by the higher structure of proteins. For example, when proteins are subjected to high-temperature, high-pressure, shear force, and other factors, the higher structure will change, thus affecting the surface hydrophobicity of proteins [12]. ANS fluorescent probe method is the most commonly used method for the determination of protein surface hydrophobicity because it has the advantages of simple operation, fast operation, and a small amount of protein. As a fluorescence probe sensitive to polarity, 8-aniline-1-naphthalene sulfonic acid (ANS) can combine with the hydrophobic cavity of protein molecules to enhance the fluorescence intensity of proteins, and the fluorescence intensity will increase with the addition of ANS until excessive fluorescence probe is added, so it can be used to determine the surface hydrophobicity of proteins [13]. Figure 1 shows the surface hydrophobicity of rice bran protein isolates prepared by high-temperature rice bran meal treated by steam flash under different conditions.

As can be seen from Figure 1, the surface hydrophobicity of RBPI decreased significantly (*p* < 0.05) after SFE treatment, and the maximum decrease was 87.3%, from 137.5 (RBPI) to 17.5 (rbPI-2.1-210), and the surface hydrophobicity of protein isolates was greatly affected by steam pressure and pressure holding time. This is consistent with the research results of Zhang Yanpeng et al. [14]. The decrease in surface hydrophobicity indicated that the structure and physicochemical properties of RBPI changed obviously during SFE. The RBPI of HTDDRB can be subjected to various physical and chemical effects, such as high-temperature cooking, mechanical shear force, and Maillard reaction caused by the instant release of steam. Among them, mechanical shearing can depolymerize protein aggregates, and some hydrophobic groups leak out. With the extension of pressure holding time, high-temperature cooking is strengthened, and these proteins gather again due to hydrophobic action, and the hydrophobic groups leaked out are hidden inside molecules, reducing the hydrophobic surface property of proteins [15]. In addition, previous studies have confirmed that the Maillard glycosylation reaction triggered by SFE can also significantly affect the surface hydrophobicity of RBPI [16]. Achouri et al. [17] modified 11S globulin by glycation and found that the decrease in surface hydrophobicity was related to the improvement of grafting degree. Meanwhile, glycation increased the hydroxyl group on the protein molecule, thus increasing the hydrophilicity of the protein, and some hydrophobic groups were shielded. Due to the Maillard reaction, the free amino group on the amino acid side chain of the protein binds sugar molecules, which affects the binding of protein and ANS, thus reducing the measured value of surface hydrophobicity. Gasymov et al. [18] found that protein glycosylation reduced lysine and arg in protein, thus reducing the binding site of ANS and protein. This is consistent with the results of this study. 

### 2.2. Intrinsic Viscosity Analysis

The intrinsic viscosity [η] is a physical property of polymer solution. Polymer solution concentration tends to zero specific viscosity, that is, polymer solution in infinite dilution of a single polymer molecule and the internal friction between solvent molecules; because the solution is very thin, polymer molecules are far away from each other, so the intrinsic viscosity [η] value is not affected by the solution concentration. However, [η] values are related to the molecular weight and molecular morphology of the polymer and increase with the increase in molecular weight and molecular stretching degree of the polymer.

Figure 2 shows the change in Intrinsic viscosity of RBPI from HTDDRB after SFE. As shown in Figure 2, the Intrinsic viscosity of RBPI prepared by HTDDRB after SFE increased significantly (*p* < 0.05), from 28.6 to 68.7 mL/g by 140.2%, and showed a trend of increasing with the increase in steam pressure and pressure holding time. The results indicated that the stretch degree of RBPI prepared by SFE was significantly higher than that of untreated RBPI, and the internal friction between protein and solvent molecules was increased, which was greatly affected by steam pressure and pressure holding time. This is mainly because of the process of SFE. The material is first subjected to heat treatment and the mechanical shear brought by the rapid movement of high-pressure steam. The heat treatment can make the protein partially denature and allow molecular aggregation to occur. At the same time, the mechanical shear brought by high-pressure steam can destroy the aggregation state of proteins and cause molecular stretching. The force is greatly affected by steam pressure. Therefore, the increase in steam pressure leads to an increase in Intrinsic viscosity and molecular stretching. The extension of protein molecules exposed more free amino groups, promoted the occurrence of the Maillard reaction, introduced more sugar molecules, and changed the spatial structure of proteins. Therefore, the extension of the pressure holding time is conducive to the occurrence of the Maillard reaction, which makes protein molecules more extended. In the later stage of steam flash, the material is subjected to shear action caused by the instantaneous release of high-pressure steam, which further causes the depolymerization of protein molecules, and the molecules are more extended. Thus, the Intrinsic viscosity is higher.

### 2.3. Thermal Stability (DSC) Analysis

Differential scanning calorimetry (DSC) is widely used to study the thermal stability of proteins. When heated, the hydrogen bonds inside proteins break, causing the protein molecules to stretch, a process that requires energy absorption, known as denaturation heat. Protein denaturation is manifested by molecular structure changing from folded state to unfolded state, from an ordered state to a disordered state, and from a natural state to a denatured state, and these processes are accompanied by energy changes. Thermal analysis of proteins involves the destruction of higher structures by heat and the measurement of energy changes in the process. 

Figure 3 and Table 1, respectively, show the DSC curve and thermodynamic characteristics of RBPI before and after SFE (steam pressure 2.1 MPa, holding time 210 s). It can be seen that the denaturation peak of RBPI before and after steam flash detonation is upward, indicating that the RBPI absorbs heat during the denaturation process. The denaturation peak temperatures and enthalpy changes of original RBPI and RBPI-2.1-210 were 114.2 °C and 106.7 °C, respectively, and the enthalpy changes of denaturation were 356.3 J/g and 231.4 J/g, respectively. The denaturation temperatures and enthalpy changes of the two were significantly different, indicating that the structure and composition of the two were significantly different.

Non-covalent bonds such as hydrogen bonds and hydrophobic interactions play an important role in the stability of protein molecular structure. They may be reversed bonds, which play a major role in thermal denaturation temperature and enthalpy change. The thermal denaturation temperature of plant proteins is closely related to their secondary structure and is generally determined by non-covalent bonds [19]. In this study, the denaturation temperature of RBPI-2.1-210 decreased after SFE, indicating that the thermal stability of the protein decreased, while the decrease in enthalpy change value indicated that the molecular structure of the protein was locally expanded, the hydrophobic groups were closer to the polar microenvironment, and some internal hydrogen bonds were likely to break, and the protein molecules were closer to the disordered state. Although raw materials are subjected to high-temperature cooking in the process of the steam flash explosion, the thermal denaturation is not obvious due to the very short time of action. Meanwhile, proteins are also subjected to mechanical shear caused by the instantaneous release of steam, which can damage non-covalent bonds in proteins, partially break hydrogen bonds, and reduce surface hydrophobicity (previous studies have confirmed this). At the same time, due to the introduction of sugar molecules into RBPI by the Maillard reaction, the non-covalent bond of protein molecules was damaged, which resulted in a decrease in thermal denaturation temperature. In addition, the thermal stability of proteins is also closely related to disulfide bonds in protein molecules. More disulfide bonds in protein molecules can promote a more stable conformation of proteins [20]. Kinsella et al. [21] found that the more disulfide bonds there are, the higher the thermal stability of proteins is. It can be inferred that steam flash treatment may damage the disulfide bond of RBPI to some extent and reduce thermal stability.

### 2.4. Size Exclusion Chromatography (SEC) Analysis (Relative Molecular Mass Distribution)

Figure 4 shows the effect of SFE treatment conditions on the relative molecular weight distribution and aggregation state of RBPI. The main peak locations of RBPI prepared before and after steam flash treatments were 8.27 min, 15.24 min, 17.28 min, 22.8 min, and 23.5 min, and their corresponding relative molecular weights were 654.4 kDa, 350.1 kDa, 238 kDa, respectively. The peaks with 24.9 kDa, 9.8 kDa molecular weight, and an 8.27 min retention time, correspond to the protein molecular aggregates (654.4 kDa). As can be seen from Figure 4, the peak value of protein isolate (RBPI) from rice bran meal at 8.27 min is very small, and the peak value of RBPI increases significantly after steam flash treatment, while the peak value of RBPI at other locations decreases. The results indicated that there were few macromolecular weight aggregates in the RBPI, but the macromolecular weight aggregates were formed in the RBPI after SFE.

In order to further analyze the relative molecular mass distribution changes of rice bran protein isolate, the chromatogram was fitted using Peakfit v4.12, and the peak integral area of each part was calculated. According to the molecular weight distribution of rice bran protein isolate, it was divided into FractionⅠ (>550 kDa). For FractionⅡ (550~80 kDa) and Fraction Ⅲ (<80 kDa), the effect of steam blitzing treatment on rice bran protein aggregates can be reflected through the change in peak integral area of each elution region, as shown in Table 2.

It can be seen from Table 2 that the relative content of Fraction I of isolated protein increased from 1.5 to 13.8% after SFE treatment, and the relative content of Fraction II and Fraction III decreased significantly, indicating that steam flash burst can promote the formation of soluble protein aggregates [22]. In addition, it can be seen from Table 2 that the distribution of the relative molecular weight of rice bran protein is greatly affected by the steam pressure and holding time, and Fraction II decreases from 58.1 to 43.9% at lower steam pressure (0.9 MPa), accompanied by the increase in Fraction I and Fraction III., indicating that some rice bran proteins depolymerize under mechanical shearing at lower pressure, and molecular aggregation occurs under high-temperature cooking, and with the increase in steam pressure, The relative content of small molecular weight Fraction III decreased significantly (*p* < 0.05), from 49.4 to 35.4%, and with the increase in Fraction I and Fraction III., the effect of holding time on relative molecular mass was basically the same as that of vapor pressure, indicating that higher vapor pressure and longer holding time contributed to the unfolding of proteins to form soluble protein aggregates [23]. Since the surface properties of proteins are greatly affected by their molecular aggregation state, SFE treatment can affect the surface properties of the RBPI and then affect its functional properties.

### 2.5. Mercaptol and Disulfide Bond Content

Disulfide bonds (SS) are important covalent chemical bonds that constitute the higher structure of proteins, which play an extremely important role in stabilizing the higher structure of proteins and maintaining their active functions, and disulfide bonds (SS) and sulfhydryl groups (-SH) can be converted to each other by redox [24]. Table 3 shows the contents of free sulfhydryl (SHF) and disulfide bond (SS) of different RBPI, among which the SHF and SS of the original high-temperature rice bran meal protein isolate (RBPI) were 4.38 μmol/g and 6.13 μmol/g, respectively, while the SHF of rice bran protein isolate was 5.54 μmol/g (RBPI-0.9-210), 5.63 μmol/g (RBPI-2.1-90) and 5.15 after SFE treatment under different conditions μmol/g (RBPI-2.1-210), SS 8.88 μmol/g (RBPI-0.9-210), 9.32 μmol/g (RBPI-2.1-90), and 7.46 μmol/g (RBPI-2.1-210), it can be seen that Steam Flash Explosion can significantly increase the SS and SHF of RBPI of HTDDRB, and the steam pressure and holding time have a great influence on it, and the SS and SHF are significantly increased (*p* < 0.05) under the long-term or high-pressure short-term treatment conditions, while the two are reduced to different degrees under the long-term treatment conditions of high-pressure (2.1 MPa, 210 s). 

The SS of the protein isolate of the original RBPI was higher than that of Wu Wei et al. [25] on the determination of fresh rice bran, which was mainly due to the oxidation of some sulfhydryl groups to disulfide bonds during the thermal stabilization of rice bran, which covalently crosslinked proteins to form thermal aggregates. In the process of SFE treatment under low-pressure long-term or high-pressure short-term conditions, the molecular structure of the protein can be destroyed due to the mechanical shearing action of high-pressure steam, and the peptide chain is stretched so that the sulfhydryl group buried inside the molecule is exposed, so that the SHF content increases. At the same time, the high-temperature cooking effect during SFE treatment can induce the formation of soluble aggregates containing disulfide bonds, which increases the SS content. When the steam pressure is 2.1 MPa, the holding time is extended from 90 to 210 s, which will greatly increase the heat treatment strength, and some studies have shown that [26] the heat treatment strength is too high but will cause some disulfide bonds to break into sulfhydryl groups, and at the same time, due to the Maillard reaction to form more non-disulfide bond covalent aggregates, free sulfhydryl groups can be wrapped inside the aggregates, so the SHF content is slightly reduced. 

### 2.6. Secondary Structure—Circular Dichroism (CD) Analysis

Protein secondary structure refers to the local spatial arrangement (folded and coiled form) of the main chain atoms in the polypeptide chain, that is, the conformation, excluding the conformation of the side chain part, mainly including α-helix, β folding, β-angle and irregular curl, and other structural forms. Circular dichroic spectroscopy is an effective means for the study of protein secondary structure. The asymmetric α-carbon atom of amino acids has optical activity, when the plane circularly polarized light passes, these optically active centers have different absorption intensities for their left and right circularly polarized light, resulting in an absorption difference, resulting in the amplitude difference of the polarized light vector, and the plane circularly polarized light becomes elliptically polarized light, which is the circular dichroism of proteins [27]. 

The far-ultraviolet circular dichroic spectrum (190~250 nm) mainly reflects the protein backbone conformation, which is often used in the study of protein secondary structure. According to the literature [28], the α-helical has a positive peak near 192 nm, two negative characteristic shoulder peaks at 208 nm and 222 nm, a negative peak at 216~218 nm at β-fold, a strong positive peak at 195 nm, and a positive peak near 205 nm at the β-turn. The irregular curl has a negative peak around 198 nm and a small, wide positive peak around 220 nm. Figure 5 shows the circular dichroic spectra of different rice bran protein isolates (RBPI), from which it can be seen that the peak shape of the circular bichromatogram of rice bran protein isolate obtained after Steam Flash Explosion treatment is basically unchanged, the peak position changes slightly, and the peak intensity changes greatly. There is a strong negative peak at 216~228 nm, which is the overlapping peak of α-helical and β-rotation angle, and this peak is significantly weakened after Steam Flash Explosion. The original high-temperature rice bran meal protein isolate had a negative peak at 197 nm, and the peak position shifted to 200~202 nm after Steam Flash Explosion treatment, which was closer to the irregular coiling peak position, and the intensity increased. At the same time, it was found that a weak positive peak appeared at 195 nm after the SFE, indicating the presence of β-folding. 

The relative content of the secondary structure of RBPI is shown in Table 4. It can be seen from Table 4 that after SFE, the α-helix content of isolated protein decreased significantly (*p* < 0.05), the content of random coil increased significantly (*p* < 0.05), and the contents of β-fold and β-corner remained basically unchanged, indicating that the secondary structure changed from ordered to disordered. This is consistent with the findings of Damodarand et al. [29]. The α-helix structure plays an important role in stabilizing the protein structure, so the stability of the protein isolate molecules obtained after SFE is reduced, the molecules are fully stretched, and the flexibility is increased, which contributes to the exertion of its functional properties. This is consistent with the findings of surface hydrophobicity, thermal stability (DSC), and intrinsic viscosity and also explains these changes in secondary structure. 

### 2.7. Tertiary Structure—Endogenous Fluorescence Spectroscopy

Based on the study of the thermal stability, surface hydrophobicity, and intrinsic viscosity of rice bran protein isolate, endogenous fluorescence spectroscopy was used to analyze the differences in the tertiary structure of RBPI prepared before and after SFE treatment. Taking 290 nm as the excitation wavelength, the endogenous fluorescence spectrum with tryptophan (Try) as the main emitting group can be obtained, and its peak migration can reflect the change in the polarity of the tryptophan microenvironment and then reflect the spatial conformation change in the protein (mainly characterizing the tertiary structure). The peak redshift indicates an increase in the microenvironmental polarity of the fluoroemitting group (Try) and greater exposure to solvents, while the blue shift indicates an increase in the microambient hydrophobicity of the emitting group (Try), while the decrease in fluorescence peak intensity is associated with fluorescence quenching [30]. Shutova et al. [31] found that the maximum emission peak of tryptophan residues, when exposed to the protein surface, was between 350~353 nm, and the maximum emission peak of tryptophan encapsulated in the protein was between 326~332 nm.

Figure 6 is the endogenous fluorescence spectrum of the RBPI obtained by different SFE treatments of HTDDRB, from which it can be seen that the maximum emission wavelength of the RBPI prepared from the original HTDDRB is 336 nm, and the peak position is high, indicating that the tryptophan residue of the isolated protein prepared from the original RBPI is relatively close to the hydrophilic microenvironment, and after the SFE, the maximum emission wavelength of the isolated protein undergoes different degrees of redshift, from 336 to 343, 345, 346 nm. It shows that tryptophan (Try) residues are gradually exposed on the surface of protein molecules and are in a more hydrophilic microenvironment [32], and the degree of migration is affected by the steam pressure and holding time. This is mainly caused by mechanical shearing and glycosylation during Steam Flash Explosion, which can depolymerize protein aggregates, stretch molecules, gradually expose embedded tryptophan, and covalent binding of proteins and carbohydrates can also enhance the polarity of the microenvironment [33]. At the same time, it can also be found from Figure 6 that the fluorescence intensity of RBPI is significantly reduced after a steam flash burst (*p* < 0.05), and it is greatly affected by steam pressure and holding time. This is mainly due to the fact that glycosylated proteins covalently bind carbohydrates, which can mask the generation of fluorescence and reduce the endogenous fluorescence of tryptophan, which is consistent with the results of Corzo-Martinez et al. [34]. In addition, some proteins can be reaggregated under high-temperature action, tryptophan side chains are shielded, and high-temperature action may also oxidize tryptophan, thereby reducing fluorescence intensity. In addition, the increase in vapor pressure and the extension of the holding time can make the protein molecules more stretched, and the protein molecules covalently bind more carbohydrates, so the maximum emission wavelength of tryptophan is larger, and the fluorescence intensity is lower. Wu et al. [35] found that the fluorescence intensity of soybean protein decreased after modification with 13-hydroperoxyoctadecadienoic acid, which was thought to be caused by changes in the aggregation state and tertiary structure of proteins. 

The above results show that after SFE treatment, the isolated protein molecules are more stretched, the polarity of the tryptophan microenvironment increases, the protein molecules are covalently bound to the sugar chain, and the tertiary structure is changed, which is consistent with the increase in surface hydrophobicity of the isolated proteins. 

### 2.8. Protein Particle Size Analysis

Dynamic laser scattering (DLS) was used to measure the diffusion of RBPI particles in solution as Brownian motion particles, which were converted into cumulant average particle sizes according to the Stokes-Einstein equation, as shown in Figure 7.

It can be seen from Figure 7 that the average particle size of RBPI from high-temperature rice bran meal was significantly affected by the Steam Flash Explosion treatment conditions and increased significantly with the increase in steam pressure and the extension of holding time (*p* < 0.05). The average particle size of the protein isolates was 102.8 nm (RBPI), and the average particle size of the protein isolates increased to 143.6 nm (RBPI-0.9-210), 151.5 nm (RBPI-2.1-90) and 168.7 nm (RBPI-2.1-210), with a maximum increase of 65.9%. This is consistent with the results shown by size exclusion chromatography, which once again proves that after SFE treatment, soluble macromolecular aggregates appear in the isolated proteins of high-temperature rice bran meal, resulting in increased protein solubility. Due to the change in the conformation of rice bran protein molecules by SFE, the existence form of spherical particles of protein molecules in an aqueous solution is destroyed, and at the same time, due to the Maillard reaction, the cross-linking between molecules is increased, resulting in an increase in the hydration radius of protein particles, so the average particle size of rice bran protein determined by dynamic laser scattering increases. Sun Tianying et al. [36] found that the isolated proteins of sunflowers underwent covalent crosslinking after heat treatment, forming macromolecular aggregates, and the protein particle size increased. 

### 2.9. Zeta Potential Analysis

There exists an abstract boundary within the dispersion layer of the double electric layer of protein particles. When the particles move, the ions inside the boundary move with the particles, but the ions outside the boundary do not move with the particles. This boundary is called the hydrodynamic shear layer or slip surface. The potential that exists at this boundary is called the Zeta potential. Therefore, the Zeta potential of protein can be used to reflect the surface charge of protein molecules. The Zeta potential (ζ-potential) of protein at pH 7.0 was determined by a Zeta potential measuring system, the Nano ZS90 nano-particle potential analyzer, and the results are shown in Figure 8. 

As can be seen from Figure 8, the Zeta potential of HTDDRB RBPI can be significantly increased by SFE, which is greatly affected by steam pressure and retention time. When the steam pressure was 2.1 MPa and the pressure holding time was 90 s, the Zeta potential of rice bran protein isolate increased by 19.4% to −24.6 mV (the Zeta of the original RBPI was −19.6 mV). When the pressure holding time was extended to 210 s under this pressure, However, the Zeta potential of RBPI decreased. This indicated that SFE could change the molecular structure of HTDDRB RBPI and increase the net charge on the molecular surface [37]. By studying the changes of endogenous fluorescence spectra of RBPI, it is concluded that the tertiary structure of RBPI is changed after sSFE, the protein molecules are more extended, and the charged polar amino acids such as tryptophan move to the surface of protein molecules, which increases the surface charge of proteins. 

### 2.10. Mechanism of SFE Treatment on RBPI from HTDDRB

Because rice bran protein contains more disulfide bonds, the molecular aggregation is strong, and the heat stabilization of rice bran results in excessive denaturation of proteins in HTDDRB, and the cross-linking effect of proteins is enhanced, and the solubility of proteins is reduced, which leads to difficult extraction. The protein in HTDDRB can be subjected to various physical effects, such as high-temperature cooking, rapid movement of high-pressure steam, and instantaneous release of mechanical shear force. These effects can destroy the molecular structure of rice bran protein, reduce the α-helix content, and increase the random curling content. The secondary structure of protein changes from order to disorder, and the molecule is more extended. At the same time, mechanical shearing can depolymerize the disulfide bond covalent aggregates of protein in high-temperature rice bran meal, and stretch the peptide chains, so that the sulfols buried in the molecules are exposed. Surface hydrophobic groups, tryptophan residues, and other charged groups move to the molecular surface, and the net surface charge increases. In addition, due to high-temperature cooking and surface hydrophobic action, the oxidation of exposed sulfhydryl groups can occur so that protein molecules reaggregate, the formation of large molecules containing disulphide bond soluble aggregates, the leakage of hydrophobic groups are hidden inside the molecule so that the protein surface hydrophobicity reduced, solubility increased; At the same time, due to the extension of protein molecules, more free amino groups are exposed, which increases the probability of collision with sugar molecules, promotes the occurrence of Maillard reaction, makes protein molecules introduce more sugar chains, increases hydrophilicity, and further improves the solubility of proteins.

## 3. Materials and Methods

### 3.1. Materials

HTDDRB (130 °C, 30 min after moist heat stabilization treatment, the powder is degreased) was purchased from Heilongjiang Beidahuang Rice Industry Group Co., Ltd. (Harbin, China); 5,5′-disulfide (2-nitrobenzoic acid) (DNTB), guanidine hydrochloride, sodium dodecyl sulfate (SDS), β-mercaptoethanol (Shanghai Boao Biotechnology Co., Ltd., Shanghai, China), the above are biotechnology grade.8-aniline-1-naphthalene sulfonic acid (ANS) (Amresco Co., Ltd., Solon, OH, USA); Sodium dihydrogen phosphate, disodium hydrogen phosphate, urea, ethylenediamine tetraacetic acid (EDTA) (Tianjin Bodi Chemical Co., Ltd., Tianjin, China), all the above reagents are analytically pure. High relative molecular weight standard proteins: thyroglobulin (669 kDa), ferritin (440 kDa), aldolase (158 kDa), albumin (75 kDa), ooalbumin (44 kDa), ribonuclease (13.7 kDa), (GE-Healthcare UK Limited).

### 3.2. HTDDRB Steam Flash-Explosion (SFE) Treatment 

The HTDDRB was crushed to 50 mesh particle size (VB-200, Beilite Vibration Machinery Co., Ltd., Xinxiang, China) by a dispersing machine (T25 basic, IKA Company, Staufen, Germany) and then divided and used. Take 400 g of crushed rice bran meal powder, according to the moisture content of 57%, and add a certain amount of distilled water to stir evenly into the SFE treatment chamber (QBS-8, Zhengdao Bioenergy Co., Ltd., Hebi, China), through a certain pressure of saturated steam and maintain pressure in a certain period of time, and then in a very short time (ms) steam flash treatment, collect steam flash samples and stored at −20 °C.

### 3.3. Preparation of Rice Bran Protein Isolate (RBPI) 

A certain amount of HTDDRB was weighed and added with deionized water according to the solid-liquid ratio of 1:20 (g:mL), and the pH was adjusted to 9.0 by acidity meter (PB-10, Sartorius Co., Ltd., Goettingen, Germany). The meal was put into a 45 °C water bath (HWS-26, Shanghai Yiheng Scientific Instrument Co., Ltd., Shanghai, China) and extracted for 2 h under the action of mechanical agitation. The supernatant was obtained by centrifuging with a high-speed refrigerated centrifuge (TGL-20M, Hunan Xiangyi Centrifuge Instrument Co., Ltd., Changsha, China) at 4000 r/min for 15 min. The supernatant was discarded, washed, and precipitated in deionized water 3 times and centrifuged in the same way. Finally, the solution was precipitated in deionized water, the pH was adjusted to 7.0, and RBPI was obtained after freeze-drying (PDU-1100, Riken (EYELA) Devices, Tokyo, Japan). The RBPI prepared from HTDDRB treated under different SFE conditions were expressed as RBPI-0.9-150, RBPI-0.9-210, RBPI-1.5-90, RBPI-1.5-150, RBPI-2.1-90, and RBPI-2.1-210, respectively (0.9, 1.5 and 2.1 represented the pressure of saturated steam, MPa. Steam pressure; 90, 150, 210 represented pressure holding time, s), and the protein isolate prepared from untreated HTDDRB is expressed as RBPI. Mico-Kjeldahl method [38] was used to determine the protein content in rice bran protein isolate and the protein content in rice bran meal. The protein purity was calculated according to Formula (1). Table 5 shows the purity of RBPI under different SFE treatment conditions.
(1)$$\mathrm{purity}\;\mathrm{of}\;\mathrm{RBPI}\;(\%)=\frac{\mathrm{Protein}\;\mathrm{content}\;\mathrm{in}\;\mathrm{RBPI}}{\mathrm{RBPI}\;\mathrm{content}}\times 100$$

### 3.4. Determination of Surface Hydrophobicity (H_0_) 

The surface hydrophobicity (H_0_) was determined by ANS (8-aniline-1-naphthalene sulfonic acid, molecular weight: 299.34) fluorescence probe [39]. The protein samples were dissolved in 10 mmol/L phosphate buffer (pH 7.0) at a protein concentration of 1.5% (*w*/*v*) (centrifugation 4000 r/min). Volumes of 10, 20, 30, 40, and 50 µL 1.5% protein solution were added to a plastic centrifuge tube containing 4 mL phosphate buffer solution, and 20 µL 8 mmol/L ANS storage solution was added before the test. The solution was shaken evenly and placed away from light for 8–15 min. The fluorescence intensity (FI) of the samples was detected by a fluorescence spectrophotometer (RF-5301PC, SHIMADZU Company, Shanghai, China). The excitation and emission wavelengths are 390 nm and 470 nm, respectively, and the slit widths are both 5 nm. The fluorescence intensity of the sample minus the reagent blank (phosphoric acid buffer) is the relative fluorescence intensity of the protein. The relative fluorescence intensity was used to plot the protein concentration, and the slope of the initial segment was used as the surface hydrophobicity index (H_0_) of the protein.

### 3.5. Determination of Intrinsic Viscosity 

The dilute solution viscosity measurement method was used. Each rice bran protein isolate was prepared into a dilute solution (0~0.008 g/mL) with deionized water as the control. A volume of 10 mL of dilute protein solution and deionized water of each concentration was placed in an Austenitic viscometer (type 1831, Shanghai Chemical Experimental Equipment Co., Ltd., Shanghai, China), and the outflow time t_0_ of deionized water and the outflow time t of each dilute solution were measured in a 25 °C constant temperature water bath (HWS-26, Shanghai Yiheng Scientific Instrument Co., Ltd., Shanghai, China). The relative viscosity *η_r_* and increased viscosity *η_sp_* of each dilute solution were calculated according to Formulas (2) and (3). Time measurement accuracy is controlled within ±0.01 s, each sample is measured in parallel 4 times, and the error shall not exceed 0.1 s. 

The concentration of dilute protein solution c is taken as the abscissa, and the ln*η_r_*/c and *η_sp_*/c are taken as the ordinate to create two straight lines, which are extrapolated to c→0, and the intercept is the intrinsic viscosity [*η*].
(2)$$\eta_{r}=\frac{t}{t_{0}}$$
(3)$$\eta_{sp}=\eta_{r}-1$$

### 3.6. Differential Scanning Calorimetry (DSC) Analysis 

The denaturation temperature of RBPI before and after SFE was measured by a differential scanning calorimeter (DSC-200F3, NETZSCH Instrument Company, Weimar, Germany). The temperature scanning range was 25~250 °C, and the heating rate was 5 °C/min [40].

### 3.7. Size Exclusion Chromatography (SEC) 

The protein solution concentration was 5 mg/mL, 0.22 μm cellulose filter membrane filtration, and the loading volume was 10 uL. Agilent 1260 HPLC system and ZORBAX GF-250/50 gel column (9.4 mm × 250 mm, particle size 4 μm/6 μm) were used for the analysis (Agilent Technologies, Santa Clara, CA, USA;). The eluent was 0.05 mol/L phosphoric acid buffer solution (pH 7.0, 50 mmol/L NaCl), the flow rate was 1.0 mL/min, the column temperature was 25 °C, and the wavelength of the UV detector was 280 nm.GE high molecular weight standard protein was used to make a relative molecular weight standard curve.

### 3.8. Determination of Sulfhydryl and Disulfide Bonding Content 

Refer to the methods of Zhang [41] and Mary et al. [42]. A mass of 100 mg rice bran protein isolate was accurately weighed and dissolved in 20 mL 0.1 mol/L pH 8.0 phosphate buffer (containing 1 mmol EDTA and 1% SDS), stirred at room temperature for 2 h, centrifuged at 10,000 r/min for 20 min (TGL-20M, Hunan Xiangyi Centrifuge Instrument Co., Ltd., Hunan, China), and the supernatant was used as the protein solution to be tested for use.

#### 3.8.1. Free Sulfhydryl (SH_F_) Determination

Take 3 mL of the protein solution to be tested and add the same volume of 0.1 mol/L pH 8.0 phosphate buffer (containing 1 mmol EDTA and 1% SDS) and 0.1 mL DNTB solution (39.6 mg DNTB dissolved in 10 mL 0.1 mol/L pH 8.0 phosphate buffer, the same below). The supernatant was mixed in a water bath at 25 °C for 1 h and centrifugation at 10,000 r/min for 20 min. The absorbance (A_F_) of the supernatant was measured at 412 nm with an Ultraviolet-visible Spectrophotometer (T6, Beijing Puchan Universal Instrument Co., Ltd., Beijing, China). The content of free sulfhydryl (SHF) was calculated according to Formula (4).

#### 3.8.2. Total Sulfhydryl Determination (SH_T_)

Take 1 mL protein solution to be tested, add 0.05 mL β-mercaptoethanol and 4 mL urea-guanidine hydrochloride solution (0.1 mol/L pH 8.0 phosphate buffer, including 8 mol/L urea and 5 mol/L guanidine hydrochloride), and then add 10 mL 12% TCA solution in 25 °C water bath for 1 h. After a water bath at 25 °C for 1 h, centrifuge at 5000 r/min for 10 min (TGL-20M, Hunan Xiangyi Centrifuge Instrument Co., Ltd., Changsha, China), discard the clear liquid, wash precipitate with 12% TCA to remove β-mercaptoethanol, and repeat twice. A volume of 10 mL of 0.1 mol/L pH 8.0 phosphate buffer (containing 1 mmol EDTA and 1% SDS) and 0.08 mL DNTB solution were added to the precipitation. After the solution was fully dissolved, the water bath at 25 °C for 1 h was centrifuged at 10,000 r/min for 20 min. The absorbance (*A_T_)* of the supernatant was measured at 412 nm with an Ultraviolet-visible Spectrophotometer (T6, Beijing Puchan Universal Instrument Co., Ltd., Beijing, China). The reaction solution without DNTB was used as the control. The total sulfhydryl (*SH_T_*) content and disulfide (*SS*) content were calculated according to Formula (5) and (6), respectively.
(4)$$SH_{F}(\mu \mathrm{mol}/g)=73.53\times A_{F}\times \frac{D}{C}$$
(5)$$SH_{T}(\mu \mathrm{mol}/g)=73.53\times A_{T}\times \frac{D}{C}$$
(6)$$SS(\mu \mathrm{mol}/g)=\frac{SH_{T}-SH_{F}}{2}$$

Formula: *C*—protein concentration, mg/mL; *D*—dilution factor.

### 3.9. Circular Dichroic (CD) Spectrum

According to the method of Shahabadi et al. [43], a certain amount of RBPI was accurately measured by freeze-dried samples and dissolved in 10 mmol/L pH 7.0 phosphate buffer to prepare 0.2 mg/mL protein solution. The solution was measured on a circular dichroic spectrometer (J-810, Japan Spectroscopic JASCO Corporation, Tokyo, Japan). Determination conditions: The optical diameter was 2 mm, the temperature was 25 °C, the sensitivity was 20 mdeg/cm, the scanning speed was 100 nm/min, the width was 1 nm, the scanning range was 190~255 nm, the response time was 0.5 s, the scanning was repeated 5 times to take the mean value, the pH 7.0 phosphate buffer was used as blank. The Spectra Manager driver software workstation of the machine provides the secondary structure of the protein.

### 3.10. Determination of Endogenous Fluorescence of RBPI

RBPI, RBPI-0.9-210, RBPI-2.1-90, and RBPI-2.1-210 were used as research objects to determine the endogenous fluorescence spectra of tryptophan by a fluorescence spectrophotometer (RF-5301PC, SHIMADZH Company of Japan, Kyoto, Japan). Each protein isolate was dissolved in a phosphate buffer solution (pH 7.0, 0.01 mol/L) and prepared into 0.15 mg/mL solution. The excitation wavelength is 295 nm, the scanning range of the emission spectrum is 300~400 nm, the slit width of excitation and emission is 5 nm, and the scanning speed is 10 nm/s.

### 3.11. Dynamic Laser Light Scattering (DLS) Determines Protein Particle Size

A certain amount of RBPI was accurately measured from freeze-dried samples, dissolved in 10 mmol/L pH 7.0 phosphate buffer, and filtered by 0.22 μm cellulose filter membrane to prepare 1 mg/mL protein solution. The diffusion of Brownian motion of protein particles was measured by the DLS component of a Malvern Nano ZS90 nanoparticle potential analyzer, which was converted into cumulative Z-average Size according to the Stokes-Einstein equation using DTS software [44]. The distribution characteristics of dynamic light scattering data at a 90° scattering Angle were analyzed by the cumulant analysis method.

### 3.12. Determination of Zeta Potential

Prepare 1 mg/mL protein solution with the same method as 3.11. The Zeta potential (Zeta—potential) of protein was determined by the Zeta potential measuring system of a Malvern Nano ZS90 nano size potential analyzer (Malvern Ltd., Royal Leamington Spa, UK). Electrophoretic mobility (UE) was measured using a combination of electrophoresis and laser Doppler velocimetry (laser Doppler electrophoresis), and Zeta potential was calculated based on the Henry equation.

### 3.13. Statistical Analysis

All experimental data were averaged over three replicates, and SPSS 24.0 data analysis software was used for one-way ANOVA and significance analysis, and experimental data processing and plotting were performed by Excel 2010 and OriginPro 9.0 software.

## 4. Conclusions

HTDDRB was treated by SFE, the surface hydrophobicity and thermal stability of the isolated proteins decreased significantly, the characteristic viscosity increased significantly, the α-helix content decreased, but the random curling content increased, and the secondary structure of the proteins tended to be disordered. The analysis of relative molecular mass distribution, endogenous fluorescence spectrum, free sulfhydryl and disulfide bonds, particle size, and the Zeta potential of binding proteins showed that after steam flash treatment, the disulfide bond covalent aggregates of rice bran protein isolates were depolymerized, the molecules extended, the sulfhydryl groups buried in the molecules were exposed, and the hydrophobic surface groups and tryptophan residues moved to the molecular surface. The net surface charge increases; At the same time, due to high-temperature cooking and surface hydrophobic action, some proteins aggregate, forming soluble macromolecular aggregates, hydrophobic groups leaked inside the molecules, and surface hydrophobicity decreased while protein solubility increased.

## Figures and Tables

**Figure 1 molecules-28-00643-f001:**
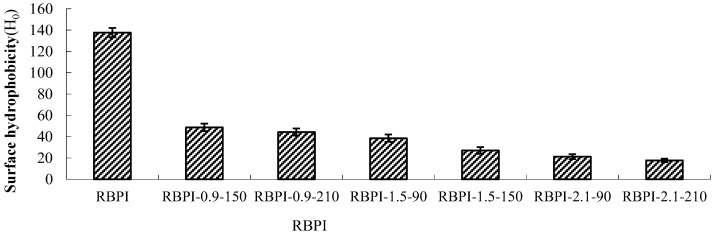
Surface hydrophobicity of RBPI prepared from different SFE pretreatment conditions.

**Figure 2 molecules-28-00643-f002:**
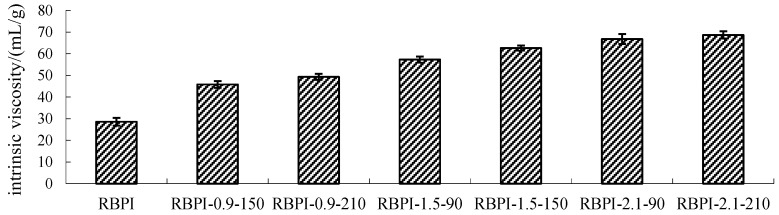
Intrinsic viscosity of RBPI prepared from different SFE pretreatment conditions.

**Figure 3 molecules-28-00643-f003:**
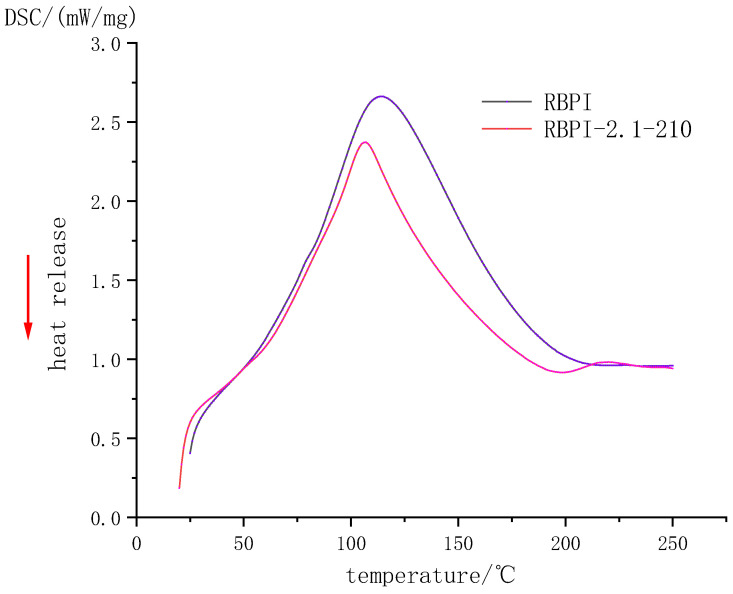
DSC Profile of rice bran protein isolate.

**Figure 4 molecules-28-00643-f004:**
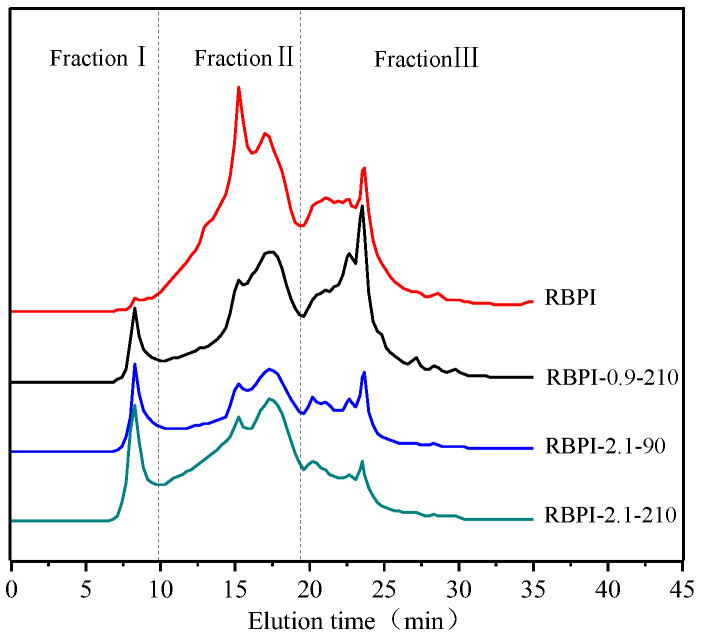
SEC profiles of RBPI prepared from different SFE pretreatment conditions.

**Figure 5 molecules-28-00643-f005:**
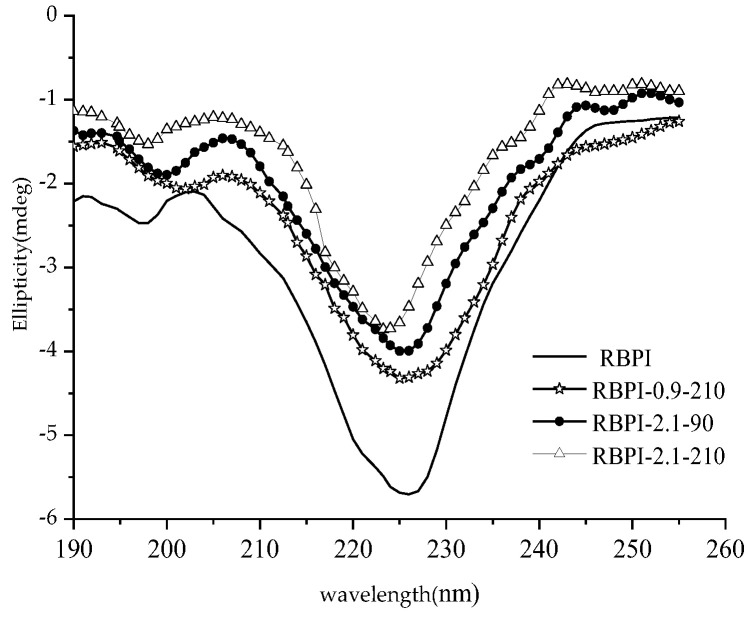
Circular dichroism spectra of RBPI prepared from different SFE pretreatment conditions.

**Figure 6 molecules-28-00643-f006:**
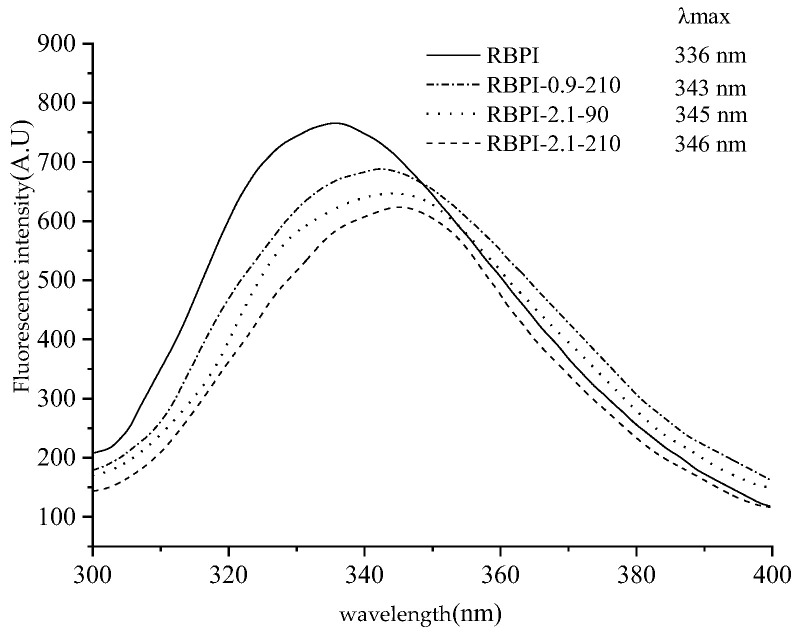
Intrinsic fluorescence spectra of RBPI prepared from different SFE pretreatment conditions.

**Figure 7 molecules-28-00643-f007:**
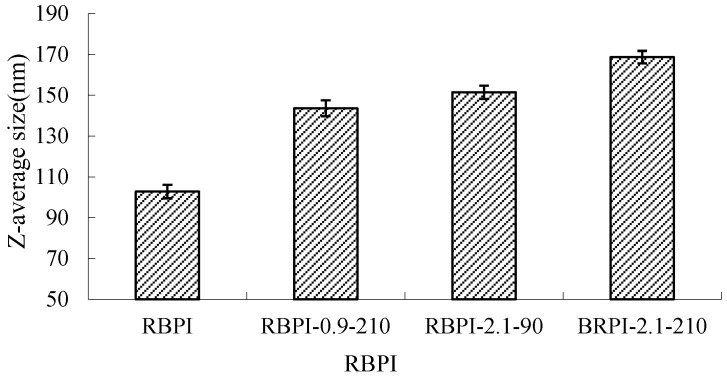
Average size of RBPI prepared from different SFE pretreatment conditions.

**Figure 8 molecules-28-00643-f008:**
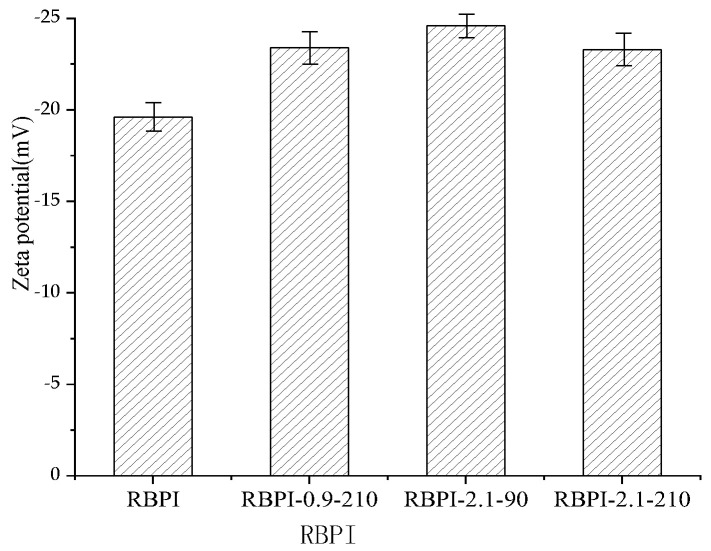
Zeta potential (ζ) of RBPI prepared from different SFE pretreatment conditions.

**Table 1 molecules-28-00643-t001:** Thermal transition characteristics of rice bran protein isolate.

Protein Sample	Starting Temperature T_o_/°C	Peak Temperature T_p_/°C	End Temperature T_e_/°C	Enthalpy ΔH (J/g)
RBPI	77.8	114.2	179.2	356.3
RBPI-2.1-210	80.7	106.7	142.6	231.4

**Table 2 molecules-28-00643-t002:** Molecular weight distribution of RBPI prepared from different SFE pretreatment conditions (%).

Elution Peak Area	RBPI	RBPI-0.9-210	RBPI-2.1-90	RBPI-2.1-210
Fraction Ⅰ (>550 kDa)	1.5 ± 0.08	6.7 ± 0.34	9.3 ± 0.16	13.8 ± 0.49
Fraction Ⅱ (550~80 kDa)	58.1 ± 1.02	43.9 ± 0.51	44.5 ± 1.03	50.8 ± 0.83
Fraction Ⅲ (<80 kDa)	40.4 ± 0.53	49.4 ± 0.24	46.2 ± 0.68	35.4 ± 0.15

**Table 3 molecules-28-00643-t003:** Free sulfhydryl and disulfide bond contents (SH_F_ and SS) of RBPI prepared from different SFE pretreatment conditions (μmol/g).

Project	RBPI	RBPI-0.9-210	RBPI-2.1-90	RBPI-2.1-210
SH_F_	4.38 ± 0.15	5.54 ± 0.08	5.63 ± 0.14	5.15 ± 0.09
SS	6.13 ± 0.16	8.88 ± 0.15	9.32 ± 0.09	7.46 ± 0.13

**Table 4 molecules-28-00643-t004:** Secondary structure content of RBPI prepared from different SFE pretreatment conditions (%).

Sample	α-Helix	β-Corner	Random Coil	β-Fold
RBPI	29.1 ± 0.25	34.1 ± 0.24	36.8 ± 0.36	0
RBPI-0.9-210	23.3 ± 0.18	33.2 ± 0.35	42.3 ± 0.14	1.2 ± 0.11
RBPI-2.1-90	21.5 ± 0.21	33.6 ± 0.17	43.6 ± 0.38	1.3 ± 0.09
RBPI-2.1-210	19.8 ± 0.09	32.4 ± 0.44	47.0 ± 0.23	0.8 ± 0.03

**Table 5 molecules-28-00643-t005:** Purity of RBPI under different SFE conditions (%).

SFE Treatment Conditions	Purity of RBPI (%)
RBPI	83.6 ± 0.86
RBPI-0.9-150	79.9 ± 0.56
RBPI-0.9-210	78.1 ± 0.64
RBPI-1.5-90	77.4 ± 0.72
RBPI-1.5-150	76.6 ± 0.66
RBPI-2.1-90	75.8 ± 0.44
RBPI-2.1-210	75.0 ± 0.78

## Data Availability

Not applicable.

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
