# Peer review of "Effect of Steam Flash-Explosion on Physicochemical Properties and Structure of High-Temperature Denatured Defatted Rice Bran Protein Isolate"

_molecules, 2023, doi:10.3390/molecules28020643_

Round 1

Reviewer 1 Report

This paper represents the good idea and has potential to be good work, but before that it neeee serious corrections. First of all there are plants technical mistakes. Abstract have to be shorter and clearer. Yo have so many repetitions of same words. Abbreviations are strongly recommended . In the section Materials and methods you didn’t use appropriate still, moreover you used more of them. Section Results also needs to be improved. The Conclusion is repeated abstract. Best regards

Author Response

请参阅附件。

Reviewer 2 Report

This study is a comprehensive investigation on the structure and physicochemical properties of rice bran meal protein isolate. The topic is worthy of publication, but the writing and the lack of method description limit the quality of the manuscript.

Major concerns

1) In abstract. “The contents of free thiol group and disulfide bond of protein isolated from rice bran meal were significantly increased at high temperature,”

This sentence in the abstract means that when authors increased treatment temperature, the free thiol and disulfide bonds of samples also increased. In other words, this sentence indicates that the treatment, or the variable in this study, is the temperature. However, reading through the maintext and the method section, it is apparent that the variables in this experimental design are pressure and time. This is an example of how English writing really limited the quality of this manuscript.  

2) The method section needs complete rewriting. The description of sample treatments and naming is supposed to be the key to this study but it is not written in a way that fits any final submission. Current wiring is more like a direct translation of a protocol rather than a completed writing for a manuscript.

3) In “3.1.1. High temperature rice bran meal Steam Flash Explosion treatment”, authors stated that samples were collected and stored at -20C. However, in “3.1.1. Preparation of rice bran protein isolate”, authors stated that they prepare protein isolates from “high-temperature rice bran meal”. How and why did authors increase the temperature of the treated meal from -20C to “high-temperature”, and what exactly is the high temperature?  If “high-temperature rice bran meal” was supposed to mean the temperature at which the meal was treated, then this is another example of how English writing limited the quality of this manuscript because readers have no way to replicate what was described here.

4) The naming of the samples was not explained at all. I can assume the numbers following RBPI means pressure and holding time, but in “3.1.1. High temperature rice bran meal Steam Flash Explosion treatment”, authors clearly stated that the meal was treated under 2.1 MPa for 210s. So, I suppose there is some other treatments during isolation step but the authors did not mention?

5) Title: “at high temperature” is somehow confusing. High temperature of the steam flash treatment, the effect is investigated at high temperature, or high temperature at which protein was isolated? Please rewrite the title

6) Table 2. Was this experiment (SEC measurement) repeated? If no, this should be repeated before drawing a conclusion. If yes, why is there no standard error?  Also, without enough data/replicates, it is not appropriate to claim on statistics like “increased significantly (P<0.05) from 1.5% to 13.8%”.

Table 4. Same concern about the replication and statistical analysis.

7) Authors measured the protein yield and isolate purity and did not mention this two throughout the manuscript. Why did authors measure these two and why did authors include them in the method description section?

8) Determination of sulfhydryl and disulfide bonding content.

Authors stated that they “Accurately weigh 100 mg of rice bran protein isolate dissolved in 20 mL of 0.1 mol/L pH 8.0 phosphate buffer, … ”.  From this description, the values calculated by the authors was based on the weight of isolate powder rather than protein weight  per sample, supposing the protein contents of each treatment are different. The comparisons of  SS and SH contents among different samples is only meaningful when the unit is based on the same amount protein (per protein weight), so this should be corrected and then the conclusions for this part can be double-checked.

Minor comments:

Abstract.

1) what is “the advanced of rice bran protein isolated structure analysis”?

2) “steam steam treatment”

3) Please rewrite “the steam steam treatment can make the rice bran protein isolated depolymerization and gathered themselves together, and the relative molecular weight distribution change is bigger, the poor solubility of small molecules of aggregate reduction associated with an increase in mac-romolecular soluble aggregates at the same time>550 kDa, the aggregate of maillard reaction products,”

4) The numbering of the materials and methods and conclusion sections should be corrected.  

5) 3.2.10. Dynamic laser light scattering (DLS) determines protein particle size. What instrument/software was used for DLS measurement? Provide experimental details like scattering angle, etc.

6) Figures.  

Figure 2 caption is copy-pasted from Fig 1.

Fig3. Indicate heat flow direction. Also, it is suggested to combine both lines into one figure. Also, for RBPI-2.1-210, there seems to be a small peak at around 220 C. What is this peak or any explanation?

Fig4. How was the molecular weight determined? This is not mentioned in method description.

7) Check the reference numbering. 

Author Response

请参阅附件。

Round 2

Reviewer 1 Report

Authors improved their manuscript and it can be published.

Reviewer 2 Report

The writing has been improved.